# A Fourier transform spectroradiometer for ground-based remote sensing of the atmospheric downwelling long-wave radiance

Giovanni Bianchini[1], Francesco Castagnoli[1], Gianluca Di Natale[1], and Luca Palchetti[1]

[1]Consiglio Nazionale delle Ricerche, Istituto Nazionale di Ottica, Via Madonna del Piano 10, 50019 Sesto Fiorentino, Italy

**Correspondence:** Giovanni Bianchini (giovanni.bianchini@ino.cnr.it)

**Abstract.** The Radiation Explorer in the Far Infrared - Prototype for Applications and Development (REFIR-PAD) is a Fourier transform spectroradiometer that has been designed to operate both from stratospheric balloon platform and from ground. It has been successfully deployed in a stratospheric balloon flight and several ground based campaigns from high altitude sites, including the current installation in the Concordia Italian-French Antarctic station. The instrument is capable to operate autonomously with only a limited need of remote control and monitoring, and is providing a multi-year dataset of spectrally resolved atmospheric downwelling radiances, measured in the 100-1500 $cm^{-1}$ spectral range with 0.4 $cm^{-1}$ resolution and a radiometric uncertainty better than 0.85 mW/($m^2$sr $cm^{-1}$).

## 1 Introduction

The measurement of the atmospheric downwelling longwave radiance (DLR) is a crucial task in climate and Earth radiation budget studies since it provides the complementary quantity to the top-of-atmosphere outgoing longwave radiance (OLR) measured from space. The knowledge of both these quantities is needed in order to achieve a complete characterization of the Earth radiation budget (ERB) (Wild, 2013).

Unfortunately, for what concerns ground-based measurements, it is very difficult to achieve a global coverage because DLR measurements can be performed only from limited locations above land areas (Ohmura et al., 1998), thus causing large errors in the estimation of the global balance of energy fluxes. This uncertainty limits our ability to identify with sufficient reliability the response (feedback) of the Earth's climate to the variation of different components (forcing) (Stephens et al., 2012).

Nevertheless, some new insights can be obtained by using spectrally resolved measurements (Huang et al., 2007; Huang, 2013). A spectrally resolved measurement of the DLR provides significant advantages with respect to spectrally integrated measurements, allowing for an accurate identification of the radiative forcing and feedback signatures, and thus the contributions to the ERB, of the various atmospheric constituents (Gero and Turner, 2011).

On the other hand, compared to standard DLR broadband integrated measurements providing the downwelling irradiance at ground level, as those coming from the Baseline Surface Radiation Network (BSRN) (Ohmura et al., 1998), spectrally resolved measurements typically measure only the radiance for a single line of sight and in a small solid angle. Further calculations, or several measurements made at different angles, are needed to estimate the irradiance. This limitation is typically present for

space measurements where the OLR irradiance is calculated from the observation of few lines of sight from polar orbit, e.g. CERES (Loeb et al., 2005, 2007), or from a single line of sight from geostationary orbit, e.g. GERB (Clerbaux et al., 2003).

This limit can be overcome with the use of a radiative transfer model and the application of an inversion procedure on the measured atmospheric emission spectra to retrieve vertical profiles of variables as water vapor, temperature and minor

constituents, which are relevant for the calculation of DLR. In practice these variables can be used in the forward model to reconstruct radiance in the lines of sight that were not directly measured (Palchetti, 2017) and thus calculating the downwelling irradiance in clear sky conditions. This approach has also been applied to satellite observations to derive CERES fluxes from IASI spectral measurements (Turner et al., 2015).

Spectral observations in the thermal infrared have been used to retrieve atmospheric state both from top of atmosphere

(TOA), see e.g Ridolfi et al. (2000) or ground-based observations (Smith et al., 1999), and to perform radiative closure experiments (Turner et al., 2004; Reichert and Sussmann, 2016). However all these observations typically cover only the Mid-IR. A few instruments have been developed to cover the far-infrared (FIR) region, defined as wavelengths greater than $15\mu$m or, approximately, above the $CO_2$ $\nu_2$ band, and are operated from ground and airborne platforms for limited timescale campaigns (Mariani et al., 2012; Green et al., 2012; Mlynczak et al., 2016).

While the relevance of the FIR spectral interval for atmospheric studies, and in particular for the study of climate, is a well-established concept (Sinha, 1995; Brindley, 1998; Harries, 2008), FIR still remains a significantly underexplored region, even more if we consider specifically long-term monitoring projects.

The Radiation Explorer in the Far Infrared - Prototype for Applications and Development (REFIR-PAD) Fourier transform spectroradiometer (FTS) has been developed with the aim of performing the spectrally resolved measurement of atmospheric

emitted radiation covering the most part of the atmospheric emission spectrum, from 7 to $100\mu$m, thus including the FIR region.

The use of room-temperature detectors and of highly reliable mechanical solutions derived from space-qualified projects (Rizzi, 2002), makes the REFIR-PAD instrument an ideal tool to perform ground-based monitoring missions on climatologically relevant timescales. This capability has been tested in 2007 with the ECOWAR campaign (Earth COoling by WAter

vapor Radiation) (Bhawar, 2008) and in 2009 with the RHUBC-II campaign (Radiative Heating in the Underexplored Bands Campaign - II) (Turner, 2012). REFIR-PAD measurement capabilities are currently being fully exploited with the installation of the instrument in the Italian-French Antarctic station Concordia, in the Dome C region on the Antarctic Plateau (75° 06' S, 123° 23' E, 3.233 m a. s. l.), where it is operating in continuous acquisition mode since December 2011.

The REFIR-PAD Antarctic campaign is performed in the framework of several research programs financed by the Italian

Antarctic Research Program (PNRA - Programma Nazionale di Ricerca in Antartide): PRANA (Proprietà Radiative del vapore Acqueo e delle Nubi in Antartide), COMPASS (COncordia Multi-Process Atmospheric StudieS), DOCTOR (DOme C Tropospheric ObserveR) and FIRCLOUDS (Far Infrared Radiative Closure Experiment For Antarctic Clouds).

Previous deployment of a FTS instrument at Dome-C dates back to the austral summer season between 2003 and 2004, when the Polar AERI (PAERI), operating in the 500-3000 $cm^{-1}$ spectral range with 1 $cm^{-1}$ resolution, was used to perform a

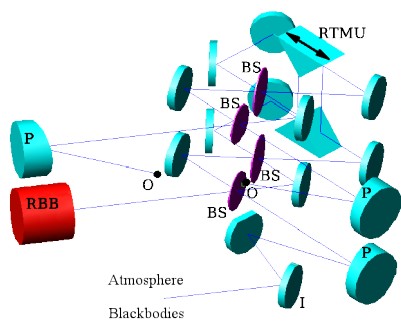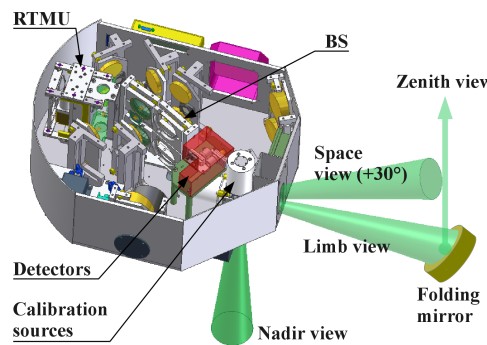

**Figure 1.** Top panel: REFIR-PAD optical layout. BS: beam splitters, P: off-axis parabolic mirrors, RBB: reference blackbody source, O: outputs (detectors), I: input selection mirror, RTMU: roof-top mirror unit (interferometric scanning mirror). Bottom panel: REFIR-PAD mechanical layout showing the actual placement of the components in the instrument enclosure and the optional folding mirror used for zenith view.

characterization of the Antarctic DLR (Walden et al., 2005, 2006). Other similar measurements were performed at South Pole (Town et al., 2005) and at Dome A (Shi et al., 2016).

In this paper a review of the main characteristics of the REFIR-PAD spectroradiometer is shown, together with the description of some measurement results obtained in ground-based campaigns in clear sky conditions; considerations and challenges related to the study of clouds are considered out of the scope of this work.

## 2 The REFIR-PAD spectroradiometer

The REFIR-PAD Fourier transform spectroradiometer is based on a Mach-Zehnder interferometer with a folded optical design that allows for a compact instrument while still retaining the moderate resolution and high throughput needed for atmospheric studies. The folding of the optical path and the number of reflections are designed to provide some degree of scanning mirror misalignment compensation (Carli, 1999a; Palchetti, 1999), allowing for a simpler mirror scanning mechanism design (Bianchini, 2006b).

The Mach-Zehnder configuration provides access to both of the two inputs and the two outputs of the interferometer, allowing for the use of a reference blackbody source (RBB in Figure 1) permanently installed on the second input. This feature, as we will see later, is critical for the reduction of beam splitter emission effects. Moreover, output separation allows to have two independent output channels.

The interferometer has the capability of operating both in a Martin-Puplett (Martin, 1969) polarizing scheme, and in a more simple amplitude-division configuration. In the first case, as shown in Figure 1, top panel, all the four beam splitters are installed, two acting as polarization divider and recombiner, and the other two, the ones nearer to the mirror scanning mechanism (Roof-Top Mirror Unit, RTMU in figure), as proper interferometric beam splitters.

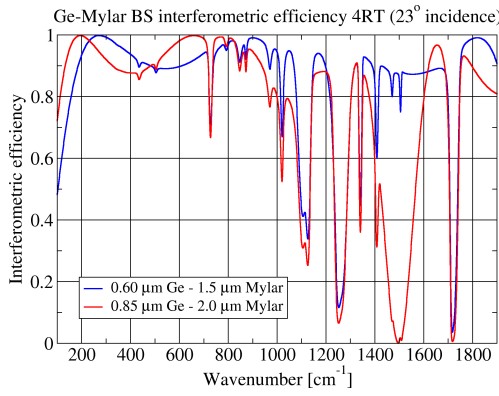

**Figure 2.** Real part of the interferometric efficiency 4RT calculated with two different configurations of the bi-layer Germanium on Mylar substrate beamsplitters.

The amplitude-division configuration makes use only of the two interferometric beam splitters, while the two other mounts are left empty. This configuration has shown to be the best choice when aiming for a wide operating spectral range, since with the use of bi-layer dielectric beam splitters the instrumental response can be tuned according to the experimental requirements.

For example, with a 0.85 $\mu$m Ge layer on a 2 $\mu$m Mylar substrate an interferometric efficiency better than 80% in the 100-1300 cm$^{-1}$ spectral range can be achieved, while with a thinner structure (0.6 $\mu$m Ge layer on a 1.5 $\mu$m Mylar substrate) the response towards higher wavenumbers can be enhanced, extending the operating range to 1900 cm$^{-1}$ at the cost of a reduction of the efficiency below 200 cm$^{-1}$ (see Figure 2). This does not constitute a problem for ground-based measurements where even in cases of extreme atmospheric transparency, with very low humidity, there is no significant atmospheric signal below 200-250 cm$^{-1}$.

In Figure 2 it is also evident that the substrate itself poses some limitations to the operating spectral range due to its absorption properties. The substrate absorption bands not only reduce the efficiency, possibly "blinding" the instrumental response as in the case of the strong features near 1250 and 1700 cm$^{-1}$, but also introduce a dephasing that makes an accurate radiometric calibration a challenging task in spectral regions close to the absorption bands (Bianchini, 2008a).

These problems could be overcome by using a different substrate, like polypropylene, which has fewer and weaker absorption bands in the region of interest, but this comes at the cost of worse optical and mechanical properties, which can critically affect the delicate process of beam splitter assembly and Germanium deposition. So Mylar has been chosen as a trade-off between theoretical efficiency and optical quality.

Problems arising from beam splitter substrate absorption, and in general from non-ideal beam splitters are also mitigated through design choices in the interferometer: the use of a reference source (RBB in Figure 1) operating at the same temperature of the instrument, and thus of the beam splitters, reduces ideally to zero the contribution to the interferogram due to beam splitter emission (Carli, 1999b; Bianchini, 2009). The orientation of the two beam splitters is also chosen in order to symmetrize the optical paths and minimize the out-of-phase contributions to the interferogram (Bianchini, 2009), as it will be discussed in

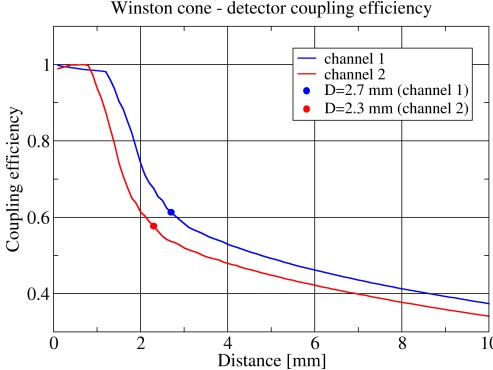

**Figure 3.** Plot of the concentrator-detector coupling efficiency as a function of their distance for both channels. The solid dots show the actual operating point, corresponding to a concentrator-detector distance of 2.7 mm for channel 1 and 2.3 mm for channel 2.

more detail in Section 8. As a matter of fact, the biggest contribution to the interferometer output due to the beam splitters in this configuration comes from the small layer thickness differences between the two beam splitters, differences that are inherent in the manufacturing process (see also Section 8).

A rotating folding mirror is placed at the instrument input port, allowing to select an atmospheric line of sight or one of the two on-board calibration sources. The rotating mirror is in the focus of a 320 mm focal length, 20° off-axis parabolic mirror that collimates input radiation towards the interferometer. The second input does not need collimating optics since its directed towards the large diameter RBB source.

The zenith line of sight that is used in the case of ground based measurements is obtained through the use of an extra folding mirror placed on the limb line of sight (see Figure 1, bottom panel).

Two 170 mm focal length, 30° off-axis parabolic mirrors focus the interferometer output ports on two 10 mm diameter Winston cone concentrators that feed the detectors. The interferometer is placed in the 1.4 m-length collimated optical path between input and output parabolic mirrors. A 22 mm pupil stop is placed in the center of the collimated path, inside the roof-top mirror unit.

The designed beam divergence $\Omega$ inside of the interferometer is 0.0027 sr, giving an instrument throughput of 0.011 cm$^2$sr. However in practice there is a limitation that is posed by the concentrator-detector coupling.

The coupling efficiency is limited by the presence of a CsI window that seals the detector case from ambient humidity. Ideally the detector should be placed as near as possible to the concentrator output aperture, but the minimum distance is actually limited by the window thickness and the distance between the window and the detector active surface.

In Figure 3 is shown the variation of the coupling efficiency with the distance between concentrator and detector. The curves corresponding to the two channels differ due to the diameter of the active surface of the two detectors (2 mm for channel 1 and 1.5 mm for channel 2). The dots show the operating condition of the two channels, corresponding to a concentrator-detector distance of 2.7 mm for channel 1 and 2.3 mm for channel 2.

As shown in Figure 3, the limitation in coupling efficiency causes a loss of about 40% in signal, but also acts as a field stop limiting the instrument field of view, reducing the beam divergence to about 0.00087 sr, for a throughput of about 0.0035 cm$^2$sr.

All the mirrors used on the REFIR-PAD instrument are coated in bare gold in order to minimize infrared absorption. Since the zenith looking folding mirror is placed outside the calibration path, its reflectivity has been characterized in laboratory and its temperature is constantly monitored in order to apply a calibration adjustment. The effect of polarization is estimated as negligible, taking into account the fact that the instrument is not operating in polarization mode and the zenith scene, in clear sky conditions, is not polarized.

Interferometric metrology is based on a paraxial laser interferometer with a 780 nm laser source (Bianchini, 2000a) that has been thoroughly tested in high-resolution FTS instruments operating both from ground (Palchetti, 2005) and from stratospheric platforms (Bianchini, 2004, 2006a).

The reference interferometer does not share any of the infrared interferometer optics, simplifying the instrument design and alignment, at the cost of having a possible misalignment between the two optical axes. This doesn't constitute a problem since it induces a linear wavenumber error which is taken care of in the wavenumber calibration procedure. This procedure, further detailed in Section 7, is based on known atmospheric line centers and does not rely on the measurement of the exact laser wavelength.

Along with the reference black body RBB, two other black body sources are used for the radiometric calibration(Bianchini, 2008a). These sources, Hot Black Body (HBB) and Cold Black Body (CBB), are placed near the instrument measurement port and can be switched into the line of sight through the rotating input bare gold mirror (see Figure 1, label "Calibration sources").

For HBB and CBB the emissivity is better than 0.999 and the operating temperature is between 10 and 80 °C. Temperature stability and temperature measurement uncertainty are both about 0.3 K, while gradients are within 0.5 K (Palchetti, 2008a).

It should be noted that limited size and good emissivity can be both achieved with these sources due to the relatively small 22 mm aperture of the blackbody which is a consequence of the placement in proximity of the focus of the input parabolic mirror.

The reference blackbody RBB has a larger diameter (64 mm) due to its placement in a collimated part of the optical path, but its requirements are more relaxed since it is not stabilized in temperature but, instead, left in thermal equilibrium with the instrument.

Acquisition of HBB and CBB radiance is performed regularly in order to obtain a constant tracking of possible instrumental response function variations. Typically a 10 minutes acquisition sequence includes 4 atmospheric measurements and 4 calibrations, 2 with HBB and 2 with CBB. Radiometric performances of the REFIR-PAD instruments are further described in section 6.

## 3   Instrumental line shape

A good model of the instrumental line shape (ILS) is a necessary requirement to correctly interpret the measured spectra and perform the level 2 data analysis (Section 10). Several effects can contribute to distort the ILS from the theoretical

$\text{sinc}(2\pi\sigma z_{\max})$ function, where $z_{\max}$ is the maximum optical path difference. Misalignment of the interferometer and scanning mirror deviations (Bianchini, 2000b) can contribute to the ILS, another possible effect is due to the finite solid angle $\Omega$ of the radiation propagating inside of the interferometer.

The effect of the finite solid angle is to broaden and shift spectral lines by convolving, in the wavenumber domain, the ideal sinc ILS with a box function extending from 0 to $\sigma_0\Omega/2\pi$, where $\sigma_0$ is the spectral line center (Vanasse, 1967).

Thus, in the optical path difference domain, the effect gives an additional, wavenumber dependent, apodization term $\text{sinc}(z\sigma_0\Omega/2)$ to be multiplied by the standard boxcar function extending from $-z_{\max}$ to $z_{\max}$. The dependency on wavenumber of the apodization function makes the exact treatment of such an effect a difficult task in the case of a broadband spectrum. A possibility is to consider $\sigma_0$ a constant, equal to the central wavenumber of the operating spectral band.

Moreover, if $\pi/\sigma_0\Omega >> z_{\max}$ the solid angle contribution to the ILS is small, and can be approximated with a triangular component in the apodization. The resulting apodization function can thus be treated as a linear combination of a boxcar and a triangle function with $\alpha$ and $1-\alpha$ coefficients, where $\alpha = \text{sinc}(z_{\max}\sigma_0\Omega/2)$.

This is a rough approximation with respect to the exact mathematical treatment of the ILS function, but since in normal instrumental operating conditions the deviations from the "ideal" ILS are very small, the effect of the approximation is negligible, and the calculation of the ILS is much faster since it makes use of the two simplest apodization functions.

REFIR-PAD ILS has been analyzed through hot blackbody calibration measurements in which isolated water vapor lines coming from residual humidity in the instrument have been identified. These features are weak enough to be far from saturation, and have a natural linewidth negligible with respect to ILS. In Figure 4 the results for two different lines are shown, one in the FIR region, at 526 cm$^{-1}$, and one at the edge of the REFIR-PAD operating region, at 1430 cm$^{-1}$. The top panels in Figure 4 (red lines) correspond to acquisitions performed with a 0.25 cm$^{-1}$ spectral sampling, the bottom panels (blue lines) to a 0.5 cm$^{-1}$ spectral sampling.

The measured lines are fitted with the $\alpha \cdot \text{sinc} + (1-\alpha) \cdot \text{sinc}^2$ approximated lineshape, corresponding to the combination of a boxcar and a triangular apodization, obtaining the corresponding value for $\alpha$.

The $\alpha$ coefficient has also been retrieved, as a function of wavenumber, for several other different spectral lines. The result of this kind of analysis is shown in Figure 5, where average $\alpha$ values are plotted vs. wavenumber. Two different series of measurements were analyzed: some performed with 0.25 cm$^{-1}$ (red circles) and others with 0.5 cm$^{-1}$ (blue squares) spectral sampling. The theoretical expression, $\alpha = \text{sinc}(z_{\max}\sigma\Omega/2)$ is also plotted, with the $\Omega$ value fitted to the experimental data.

Both the datasets provide the same $\Omega$ value, as expected, also, the fitted value (0.0008 sr) is smaller than the theoretical beam divergence given by the optical design (about 0.0027 sr), but in good agreement with the actual value of 0.00087 sr calculated taking into account the limitations in coupling efficiency due to the finite distance between Winston cones output aperture and detectors (see Section 2).

It should be noted that at low wavenumbers the solid angle contribution is completely negligible, but even in this case the line fitting gives an $\alpha$ value lower than 1 (typically about 0.95). This can be explained with the fact that there are other contributions to the ILS (residual misalignment, optics planarity, scanning mirror movement irregularities) that can give a residual contribution that is visible when the solid angle effect is negligible.

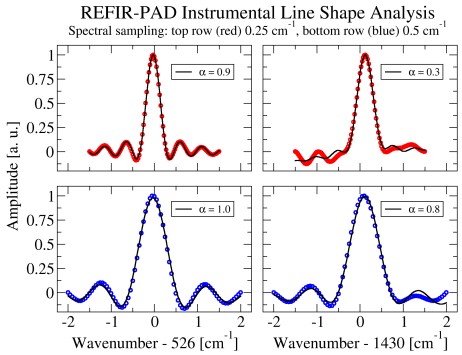

**Figure 4.** REFIR-PAD instrumental line shape. Blue line shows the isolated atmospheric line used for the analysis, the instrumental line shape is a linear combination of $\mathrm{sinc}$ and $\mathrm{sinc}^2$ components.

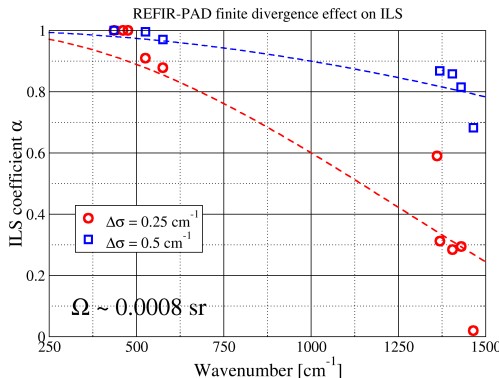

**Figure 5.** REFIR-PAD instrumental line shape coefficient obtained through analysis of measured spectra for 0.25 and 0.5 $cm^{-1}$ nominal resolution. Continuous lines show the $\mathrm{sinc}(z_{\max}\sigma_0\Omega/2)$ theoretical behaviour.

We also observe that, since in case of a small amount of interferometric misalignment the effect on the ILS can be approximated at the first order with an increase of the $\mathrm{sinc}^2$ component, it is possible to treat the interferometric misalignment in level 2 data analysis through fitting the $\mathrm{sinc}/\mathrm{sinc}^2$ ratio as an extra parameter (see Section 10). This is a very useful feature in the case of remote operation in extreme environments, an operating condition in which a slight misalignment is always a possibility.

## 4  Detectors and data acquisition electronics

One of the defining characteristics of the REFIR-PAD spectroradiometer is the use of room-temperature detectors to cover the middle to far-infrared spectral range. This result is obtained through the use of high-sensitivity Deuterated L-Alanine doped

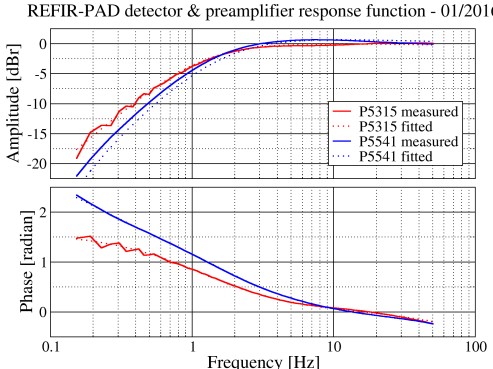

**Figure 6.** Top panel: REFIR-PAD detector preamplifier response as measured in normal operating conditions, along with a fit of the theoretical model used to measure the detectors characteristic low and high pass frequencies.

Triglycine Sulphate pyroelectric detectors provided by BAE-Selex (models P5315 and P5541). Specifications for the P5315 (P5541) at $f = 100$ Hz are: Detectivity $D^* = 5.0 \cdot 10^8 (5.3 \cdot 10^8)$ cm$\sqrt{\text{Hz}}$/W, Responsivity 1250 (450) V/W.

The detector active area diameter is 2 mm for P5315 and 1.5 mm for P5541. To enhance the light-gathering ability of the detectors, Winston cone concentrators are mounted in front of them (see Section 2).

The detectors are specified for 10-3000 Hz operating frequency range. In standard operating conditions ($3.3 \cdot 10^{-2}$ cm/s OPD scanning speed, 100-1500 cm$^{-1}$ spectral range) the REFIR-PAD instrument operates in the 3.3-49.5 Hz frequency interval. This is partially outside of the low end of the specified operating range, thus an accurate characterization of the detector system is required.

    The typical frequency response of a pyroelectric detector is characterized by "crossed" low and high cutoffs resulting in

a strongly frequency-dependent amplitude and phase. The presence of a low frequency cutoff is rather an advantage in an intrinsically AC-coupled application like FT spectroscopy, but, on the other side, a frequency-dependent dephasing constitutes a severe problem in a FT spectrometer, and must be solved by the use of a specifically designed preamplifier with a tailored response function in order to obtain a flat response and a very low dephasing across the operating frequency range.

    In Figure 6 the response of the detector and preamplifier is measured in operating conditions, supplying to the detector an

optical step function through the use of a laser and a shutter. The resulting response function can be fitted with a mathematical model of the detectors two-pole response multiplied by the preamplifier electronics response in order to obtain an estimate of the actual frequencies of the detector poles. The fitted function is shown in figure as a dotted line, and is in a very good agreement with the measured data.

    The values for the low and high frequency cutoffs obtained by the fitting process shown in Figure 6 are used, together with

the mathematical model of the preamplifier response, to provide an estimate of the residual dephasing to be used in the phase correction algorithm in the level 1 processing of the interferograms (Bianchini, 2008a).

# 5 Control and data storage system

The REFIR-PAD instrument features an on-board control unit that allows for autonomous operation (shown as a purple box in the mechanical layout shown in Figure 1).

The on-board control unit is based on a PC-104 industrial computer with a 486DX2 CPU operating at 100MHz and 32 MB of RAM. Storage is provided by a 64 GB SSD or, in alternative, a removable CompactFlash card slot (the use of which bypasses the SSD).

The on-board unit runs a streamlined version of Debian GNU/Linux v. 3.0 in which all the non-essential services have been disabled to reduce system disk access and thus increase robustness in case of loss of power. At system boot, after enabling networking, the REFIR-PAD control program is launched, immediately starting the data acquisition sequence.

This setup allows for compactness and robustness, and is ideal for the balloon-borne operation mode and for short ground-based campaigns: the SSD can store up to a month of continuous measurements, while the CompactFlash slot allows to easily retrieve data after a measurement run.

The permanent installation at Concordia station has instead required an upgrade of the control systems surrounding the REFIR-PAD instrument in order to provide continuous, unattended operation capabilities.

At Concordia station the REFIR-PAD instrument is installed indoors, in a shelter near the main base, enclosed in a thermally insulated box that is connected to an opening on the shelter roof by means of an insulated chimney. In this way, even if no window is used to separate the instrument from the outside environment, the shelter inside is kept protected from the outside air. The measurement port on the roof of the shelter can be closed by means of a motorized door when the instrument is not operated (and closes automatically in case of loss of power).

An autonomous microcontroller-based thermal control system is used to keep the instrument at a constant temperature (within $+/- 0.5$ K) through a set of heaters and a fan-driven inlet tube extracting cool air from the bottom of the shelter. The thermal control system is also provided an Ethernet connection and can be remotely controlled and configured through a minimal web interface.

The REFIR-PAD FTS is remotely operated through a second computer placed in the shelter and connected with a direct point-to-point Ethernet link to the FTS on-board control unit. This control and storage computer can switch on power to the FTS, to the view port door and to the heating system. It does also share, through the NFS protocol, a 2 TB RAID-1 disk array which is mounted at boot by the REFIR-PAD on-board control unit in order to store the acquired data.

The normal operation sequence of this setup consists in the control and storage computer opening the measurement port, turning on REFIR-PAD and waiting for a preset interval (typically configured as about 5-6 h).

At the end of the measurement run, it shuts down REFIR-PAD through the network and proceeds to compress and archive the raw data in a time-stamped directory structure and finally to perform level 1 pre-processing.

This pre-processing step is needed in order to send to Italy the calibrated and averaged DLR spectra since the full amount of raw data produced by the FTS is too large for a direct transfer.

After the end of the data pre-processing, a new measurement run is started.

Together with REFIR-PAD data, the control and storage computer performs acquisition and storage of several auxiliary parameters ranging from weather parameters outside the shelter measured by a Vaisala WXT520 station to diagnostic temperature values coming from different sensors placed in the shelter and inside of the instrument box.

The control and storage computer is always on and can be accessed remotely, allowing for the complete control of all the acquisition parameters even when REFIR-PAD is not operating, since the configuration files reside on the RAID-1 disk array.

Advantages of this architecture are redundancy and fail-safe operation: in case of a malfunction of the control and storage unit the REFIR-PAD instrument can still operate autonomously within the 1 month data storage autonomy provided by the on-board SSD. The control and storage unit can be easily replaced with a pre-configured, identical, spare unit available in the shelter. On the other side, in case of a malfunction of the REFIR-PAD on-board unit, a serial console active at boot time and accessible from the control and storage unit allows for remote troubleshooting including Power-on Self Test monitoring and BIOS configuration.

The raw data amount produced by the REFIR-PAD instrument is about 40 GB/day (2 GB/day compressed), the corresponding level 1 preprocessing output is about 50 MB/day (12 MB/day compressed), an amount of data which can be easily transferred even with the low-bandwidth connection provided at Concordia station (512 kb/s maximum).

## 6   Radiometric performances

A direct estimate of the radiometric accuracy of the REFIR-PAD spectra can be obtained through the signal measured in a spectral interval in which complete atmospheric transparency is expected. In the case of a high-altitude, extremely dry environment, this condition is achieved in the atmospheric window around 800-1000 cm$^{-1}$.

Specifically we used the dataset acquired from the Cerro Toco site at about 5500 m a.s.l. in the Atacama region, Chile, during the RHUBC-II campaign (Turner, 2012). Of this dataset, we selected only measurements that have a PWV lower than 0.6 mm. In these cases the expected atmospheric radiance signal in a narrow interval between 828 and 839 cm$^{-1}$ is completely negligible, so this spectral region can be effectively used to check the instrument radiometric accuracy in the middle of its operating band.

In Figure 7 a statistical analysis of the distribution of the average radiance in the selected interval is presented. The distribution has been fitted with a Gaussian curve, obtaining a negligible offset and a standard deviation of about 0.7 mW/(m$^2$sr cm$^{-1}$). The latter is in a good agreement with the *a-priori* estimate of the radiometric error obtained combining the noise equivalent spectral radiance (NESR) and the calibration error (Bianchini, 2008a): the estimated NESR in the selected spectral band is <0.6 mW/(m$^2$sr cm$^{-1}$) and the calibration uncertainty is 0.6 mW/(m$^2$sr cm$^{-1}$), giving a total uncertainty (through root sum of squares) of about 0.85 mW/(m$^2$sr cm$^{-1}$).

The much lower value of constant bias shows that systematic errors in the calibration procedure are negligible with respect to the estimated radiometric uncertainty.

This method for checking the calibration accuracy doesn't provide a characterization through the whole spectral range, but can be performed whenever needed, during a multi-year deployment of the instrument, provided very dry atmospheric

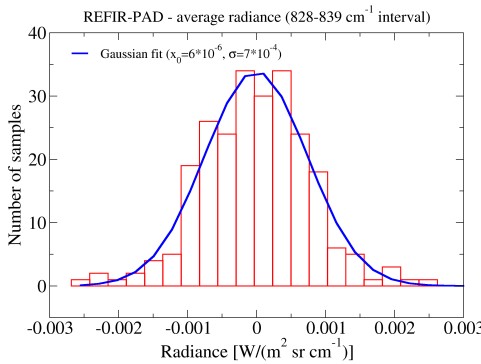

**Figure 7.** Statistical analysis of the average radiometric signal in a high transparency window (828-839 cm$^{-1}$) for RHUBC-II measurements. Radiometric bias is closed to zero, while the half width of the Gaussian distribution is 0.7 mW/(m$^2$sr cm$^{-1}$).

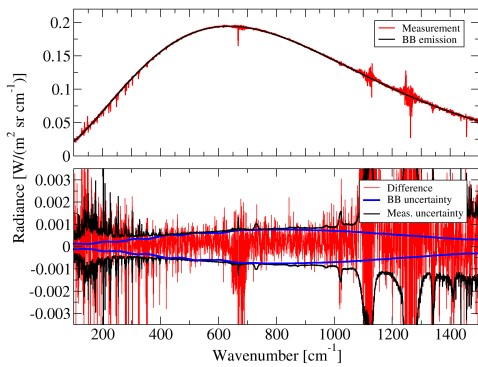

**Figure 8.** Measurement of a reference calibration blackbody sources at about 324 K.

conditions are present (which is frequently the case for Dome C). An estimate of the calibration error covering the full spectral range can instead be obtained only with a dedicated measurement performed using an external reference blackbody (BB) placed on the instrument measurement port. Figure 8 shows the results of such a calibration measurement (red line in the top panel) compared with the reference BB calculated emission (black line in the top panel). In the bottom panel, the difference

5 (gray line) is compared with the estimated measurement uncertainty (black line) and the BB calculated emission uncertainty (blue line).

The results show that the calibration accuracy is quite constant over the 300-1000 cm$^{-1}$ spectral range. Below 300 cm$^{-1}$, and in correspondence of the beam splitter substrate absorption bands above 1000 cm$^{-1}$, the measurement errors are prevalent and it is difficult to quantify the actual calibration accuracy.

## 7 Spectral calibration

The use of a diode laser as a metrology source for the REFIR-PAD spectroradiometer has allowed for increased ruggedness and compactness of the system; however, this is at the cost of a lower absolute stability of the spectral calibration reference.

The wavelength of a diode laser emission depends strongly on both the diode temperature and drive current. The unit used in REFIR-PAD features a specifically designed control unit providing both temperature stabilization and a high stability, low noise current drive. Besides temperature and drive current, diode lasers typically feature a large device-to-device wavelength variability. For this reason, once the reference source is installed and set up, its wavelength is calibrated and stored as a level 1 analysis software configuration parameter.

The main source of laser frequency error is due to thermal drifts in the temperature and current control and has been evaluated from the electronic component specifications in contributions of about 120 MHz/K (0.31 ppm/K) from laser current and 60 MHz/K (0.16 ppm/K) from laser temperature.

Assuming a maximum temperature fluctuation of 2 K, a safe estimate considering the performances of the instrument temperature control subsystem (see Section 5), we obtain a laser frequency error better than 1 ppm. This, in normal operation, allows for the use of a single frequency calibration even in case of long term measurements.

The observed laser frequency drift (see Section 11) is about <15 ppm/year, still low enough to allow to perform frequency calibrations monthly or even yearly. Nevertheless, a more robust automatic frequency calibration procedure has been developed to treat specific cases in which the above mentioned frequency stability cannot be reached, e.g. in case of laser mode jumps, or large temperature drifts of the instrument environment.

The procedure is based on the line fitting of the residual absorption due to the $CO_2$ $\nu_2$ band that is observed in the hot black-body calibration measurements. This approach has been chosen in order to have a reference spectrum that is as independent as possible from the measurement conditions, so the calibration procedure does not need to be adjusted according to the observed scene. Also, the absorption spectrum used in the procedure can be simply modeled using line strengths and the $\mathrm{sinc} + \mathrm{sinc}^2$ instrumental line shape (see Section 3). The downside of this approach is that a per-spectrum calibration cannot be performed, since a calibration coefficient is obtained only from the hot blackbody measurements. This does not constitute a limitation as long as the laser frequency drifts are negligible on the timescale of the calibration measurements repetition rate (about 10 minutes). In Figure 9 a sample result of the fitting process is shown.

Since the frequency fluctuation that a diode laser can experience can be quite large, a two step fitting procedure has been developed. First a simple peak finding algorithm is applied on the Q branch of the $\nu_2$ band, then the fitting of the P branch in the 635-665 cm$^{-1}$ spectral region is performed, as shown in Figure 9. The first stage of the process prevents, in case of a frequency drift that is larger than the P branch line spacing, a systematic error in the second stage due to the periodic structure of the spectrum.

The last step needed to perform the frequency calibration of the atmospheric measurements involves a linear regression in time of the frequency shift coefficients obtained from the hot blackbody measurements, the result of which is used to calculate the frequency drift correction for each of the atmospheric measurement.

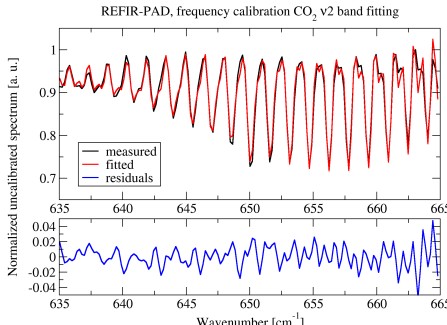

**Figure 9.** Result of the line fitting process used to perform automatic frequency calibration of the REFIR-PAD measured spectra.

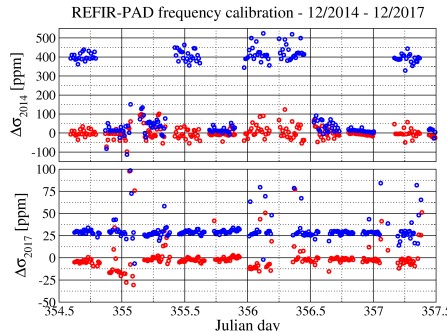

**Figure 10.** Effect of the automatic frequency calibration procedure. Blue circles represent the frequency correction factor retrieved by the level2 data analysis process with uncorrected data, red circles show the same parameter retrieved from automatically frequency-calibrated spectra.

In Figure 10 it is shown the effect of the automatic frequency calibration procedure in two different case studies. In both cases the frequency shift retrieved by the level 2 data analysis (see Section 10) is plotted with standard (blue curve) and automatic (red curve) frequency calibration.

During December 2014 (top panel) the reference laser showed bistable operation due to operating parameters being near to a mode jump. This caused the laser to operate on randomly one or the other of the two nearby modes. The separation between modes corresponds to a 400 ppm frequency shift (blue curve). After automatic frequency calibration it can be seen that the frequency shift coefficient variability (red curve) is reduced to the effect of measurement noise on level 2 analysis.

In bottom panel it is shown the same method applied to a data set from December 2017, at the end of the 2 year period shown in Figure 18. While the overall effect observed is a general reduction of 25 ppm offset, it should be noted that the automatic calibration procedure itself induces fluctuations that can be as large as 10 ppm due to the accuracy of the fitting process.

# 8 Instrument mathematical modeling

A simulation software has been developed with the intent of providing a tool to estimate the expected performance of the REFIR-PAD interferometer. The code is written in MATLAB-compatible language, and takes into account all relevant elements of the instrument geometry and optical design. The main scope of this tool is to assist in the design and test of the beam splitters, and to identify and verify the configuration providing the best optical path compensation.

The simulation assumes a generic Mach-Zehnder design with two independent inputs and two outputs. The beam splitters are modeled as asymmetric multilayers characterized by generally different optical reflectivities for the two sides $R_1$, $R_2$.

In the simulation the two inputs can be associated to a blackbody source or to a synthetic spectrum provided by an atmospheric forward model, so in both cases a real function. The two sources are split according to the calculated complex transmission and reflection coefficients of the first beam splitter as obtained by the dielectric multilayer theory using the measured complex refraction indexes for the different layers in order to correctly represent bulk material absorption.

The emission of the first beam splitter, due to its non-null absorption, is also considered as an independent source. Beam splitter absorption is mainly caused by the substrate absorption bands, so it appears as a localized and easily identifiable effect.

The two arms of the interferometer are then recombined on the second beam splitter whose properties are calculated in the same way as for the first.

Effect of the misalignment of interferometric components is calculated in the circular beam approximation using Bessel functions, planarity error is also modeled using simple approximations (spherical or trapezoidal deformation) and integrated on the beam profile.

The resulting complex spectrum is multiplied for a real absorption spectrum simulating the effect of air inside of the instrument (most of the absorption takes place outside of the interferometric path thus it doesn't produce dephasing) and the effect of the detector windows, obtaining the total signal incoming on both detectors. An inverse Fourier transform is applied to this signal to generate the simulated interferogram, which is then processed with the standard level1 data analysis chain used to process the REFIR-PAD measurements and described in detail in Bianchini (2008a).

In Figure 11 a comparison between measurements and simulation of an acquisition of the internal hot blackbody source is shown. The four top panels show amplitude and phase of the signals observed on the two output channels in case of a setup in which the beam splitter coated surfaces are facing opposite directions. Red lines correspond to measured spectra and blue line to simulations. The bottom four panels show the same signals obtained in a configuration in which the beam splitter coated surfaces are facing the same way, i.e. the configuration occurring in the case a single, homogeneous, beam splitter surface is used both as beam divider and recombiner in the Mach-Zehnder interferometer. The tests were performed with the 0.85 $\mu$m Ge on 2.0 $\mu$m Mylar beamsplitter design.

It appears evident that the second configuration does not allow for a good compensation of the optical paths, as can be seen by the large oscillations in the phases. This improves greatly with the use of opposite-facing beamsplitters. It should be noted that, as the model confirms, the residual phase undulations that are still observed in the latter configuration come from small differences, of the order of few tens of nm, in the thickness of the layers composing the two beam splitters.

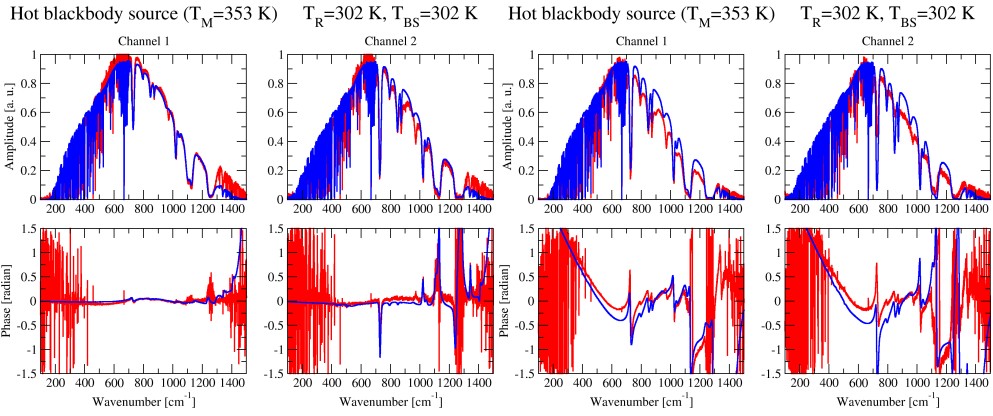

**Figure 11.** Simulation of the REFIR-PAD instrumental response function for different beamsplitter setups (blue line). Compensated (top 4 panels) and unbalanced (bottom 4 panels) configurations are shown. Laboratory measurements (red line) are compared with simulation outputs for the corresponding configuration.

On the other side, the sharp peaks observed around 700 cm$^{-1}$ and above 1000 cm$^{-1}$ originate from absorption bands in the Mylar substrate and impact both in the amplitude and phase of the measurements.

The minimum in interferometric efficiency near 1500 cm$^{-1}$ that is due to the periodical characteristic of multilayer beam splitters is also correctly modeled. Actually the minimum appears split in two due to the small differences in thickness between the two beam splitters mentioned above.

The opposite-facing beamsplitter configuration, providing the best optical path compensation, is the one used in the REFIR-PAD instrument. This configuration gives, for each interferometer input, an output with almost complete compensation (channel 1 in figure 11) and one with at least partial compensation.

## 9 Level 1 products

The main data product of the REFIR-PAD spectroradiometer is the calibrated atmospheric emitted radiance integrated in the field of view of the instrument (a cone with an aperture of about 10°) and spectrally resolved with a 0.4 cm$^{-1}$ resolution in the 100-1500 cm$^{-1}$ range. The calibration procedure, described in detail in Bianchini (2008a) follows the complex calibration described by Revercomb et al. (1988). The onboard reference blackbody sources are simulated using a specific mathematical model shown in Palchetti (2008a).

Figure 12 shows a typical calibrated spectrum of the DLR (red line) acquired at Dome-C with an integration time of 5 min. The residual imaginary part of the spectrum, after calibration, is also shown (blue line), which is comparable with the estimated noise (black line) as expected.

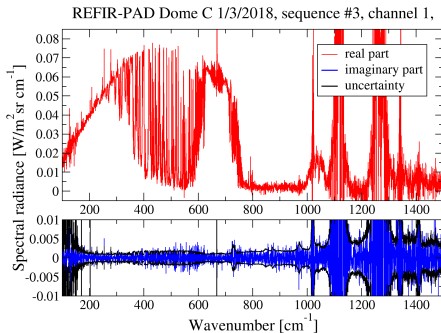

**Figure 12.** Real (red line), imaginary (blue line) and noise (black) for a typical zenith-looking calibrated spectrum.

|  | Date | Duration | Int. time | Rep. rate | Spectral range | Resolution |
|---|---|---|---|---|---|---|
|  |  | (UTC) | (min) | (min) | (cm$^{-1}$) | (cm$^{-1}$) |
| Teresina, Brazil | 30 June 2005 | 8:05–15:48 | 6.4 | 10.4 | 100–1100 | 0.475 |
| Monte Morello, Italy | 6 February 2006 | 16:26–17:58 | 5.1 | 7.7 | 350–850 | 0.5 |
| Monte Gomito, Italy | 13 – 14 March 2006 | 16:20–9:30 (+1) | 6.1/9.9 | 9.2/15.7 | 350–1100 | 0.5 |
| Testa Grigia, Italy | 4 – 13 March 2007 | 6 days | 5.1 | 11.0 | 240–1400 | 0.5 |
| Breuil-Cervinia, Italy | 15 March 2007 | 15:14–23:09 | 5.1 | 11.0 | 350–1400 | 0.5 |
| Pagosa Springs, USA | 22 – 29 April 2009 | 6 days | 5.1 | 11.0 | 350–1400 | 0.5 |
| Cerro Toco, Chile | 21 Aug. – 24 Oct. 2009 | 37 days | 5.1 | 11.0 | 100–1500 | 0.5 |
| Testa Grigia, Italy | 9 – 11 March 2011 | 3 days | 5.1 | 11.0 | 240–1400 | 0.25 |
| Dome C, Antarctica | since 21 December 2011 | permanent | 6.4/5.5 | 14.1/11.9 | 100–1500 | 0.4 |

**Table 1.** Data available from the measurement campaigns performed by the REFIR-PAD instrument. Integration time corresponds to the actual acquisition time in zenith looking mode used to produce a single spectrum, while the repetition rate accounts for the total duration of the acquisition including calibrations and system overhead.

The REFIR-PAD spectroradiometer has been operated in several campaigns in different environments (tropical, midlatitude, polar), and at different working altitudes, from about sea level to over 5000 m a.s.l. (Bianchini, 2007; Bhawar, 2008; Turner, 2012). Table 1 shows a list of the campaigns with some information about the available datasets. Of particular importance is the dataset acquired at Dome-C, Antarctica (about 3200 m a.s.l), where the instrument has been acquiring spectrally resolved DLR in all-sky conditions since the end of 2011 (Palchetti, 2015).

In Figure 13, top panel, a set of calibrated spectra acquired in different atmospheric conditions is shown. Each spectrum correspond to an average of about 6 h of measurement in clear sky conditions. The measurements span about 2 orders of magnitude in terms of atmospheric total precipitable water vapor (PWV).

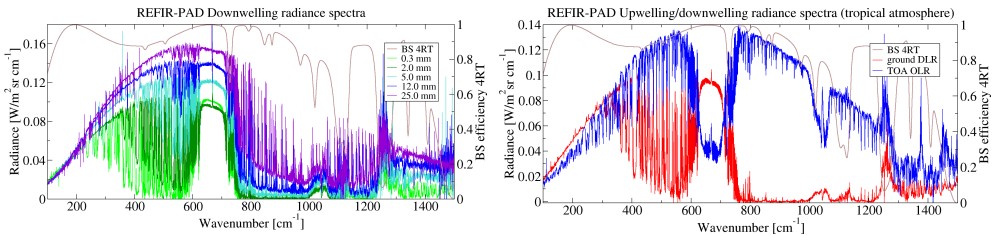

**Figure 13.** REFIR-PAD level 1 data products. Top panel: average zenith-looking spectra corresponding to about 6 h of data acquisition obtained in different atmospheric humidity conditions, spanning about two order of magnitude in terms of total precipitable water vapor (PWV). Beam splitter efficiency curve is also shown, to explain noise bands. Bottom panel: high-altitude ground-based zenith-looking spectrum (Cerro Toco, Chile) compared with top of atmosphere, nadir-looking measurement from stratospherc balloon (Teresina, Brasil).

The REFIR-PAD instrument has also been operated in nadir-looking observation mode from stratospheric balloon platform (Palchetti, 2006), obtaining atmospheric emission spectra from a 38 km altitude, thus comparable, for practical purposes, to the TOA condition. In Figure 13, bottom panel is shown a comparison between the TOA spectrum acquired during the flight and a ground-based zenith-looking measurement performed during the RHUBC-II campaign (Turner, 2012).

The measured spectral range includes almost all of the thermal emission from the Earth's atmosphere. If we consider the far-infrared region (200-667 cm$^{-1}$), that is the main scientific target of the REFIR-PAD instrument, using the radiometric accuracy figures provided in Section 6 and the spectra shown in Figure 13, we obtain a relative uncertainty in the measurement of the total radiance which lies between 0.7% and 2%.

## 10   Level 2 products

The REFIR-PAD level 1 data products can provide plenty of information not only on the radiative properties of the atmosphere, but also on its structure and composition. To perform the retrieval of these variables, a software package has been developed (Bianchini, 2011) that is based on the Line-By-Line Radiative Transfer Model (LBLRTM) (Clough, 2005) and the MINUIT minimization routines, part of the CERNlibs.

The software retrieved temperature and water vapor content profiles on separate vertical grids, together with extra parameters
like columnar amounts of minor species, cloud optical thickness and instrumental parameters as wavenumber calibration shift and line shape coefficient $\alpha$. The spectral range used in the process is a subset of the full REFIR-PAD spectral range, typically 350-850 cm$^{-1}$, even if some adjustments to the low-wavenumber end can be made according to the observed PWV ranges.

In detail the retrieval code makes use of the subroutine MIGRAD, which is based on the Davidon-Fletcher-Powell (DFP) algorithm, to minimize the chi square cost function given by:

$$\chi^2 = (\boldsymbol{y} - \mathbf{F}(\boldsymbol{x}))^T \mathbf{S}_y^{-1} (\boldsymbol{y} - \mathbf{F}(\boldsymbol{x})), \tag{1}$$

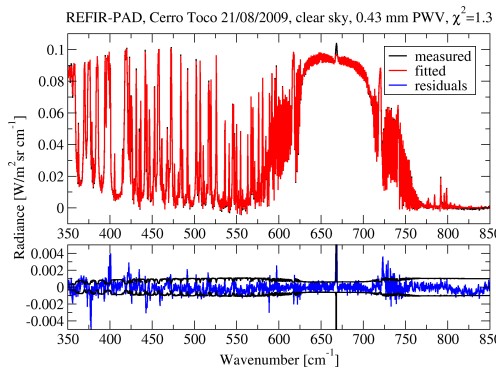

**Figure 14.** A typical result of the level 2 data analysis process. A single REFIR-PAD measurement is fitted using the LBLRTM forward model and the MINUIT minimization routines. Fitting residuals are compared with the total radiometric uncertainty (black line) in the bottom panel.

where $y$ and $x$ are the vector of the measurements and the state of the atmosphere respectively, $\mathbf{F}$ is the forward model (LBLRTM version 12.2 in our case) and $\mathbf{S}_y$ is the diagonal variance-covariance matrix for the measurements. The DFP algorithm is a quasi-Newton method which does not require the calculation of the jacobians at each iteration but uses an approximated form. This algorithm updates the inverse hessian matrix calculating the derivatives just at the first step and then using

the iterative formula shown above. The same fitting approach which was applied in a previous works (Bianchini, 2011), was used in this paper. No *a-priori* information was assumed as regularly done in a Bayesian approach, such as optimal estimation, and the initial guess is represented by a local monthly climatology, obtained averaging over a set of radiosoundings daily performed at Dome-C. Since no *a-priori* information was used to constrain the solution and no regularization was introduced, to avoid the oscillation effects due to the ill-conditioning of the problem this approach requires to limit the number of retrieved

parameters, hence the number of fitted levels both for water vapor and temperature profiles is equal to the number of degrees of freedom (DOF). The DOF were derived from a preliminary study performed through singular values decomposition of the Hessian matrix which includes Jacobian and the measurements noise.

A typical retrieval operates on a 4-point grid for temperature and a 5-point grid for water vapor, with retrieval levels chosen on the base of an analysis of the Jacobians of the selected variables. In Figure 14 the typical result of a fitting process is shown.

Even in this case adjustments to the retrieval grid can be made according to the observed atmosphere properties: as an example, in case of Antarctic measurements, the high atmospheric transparency and peculiar vertical structure (coming also from the perturbating effect given by the presence of the shelter in which the instrument is installed) allow for a retrieval of a 5-point profile also for temperature.

The vertical profiles of temperature and humidity obtained from the analysis of the set of zenith-looking measurement shown

in Figure 13, top panel, are presented in Figure 15. These results show how the process can operate in a very wide range of atmospheric conditions.

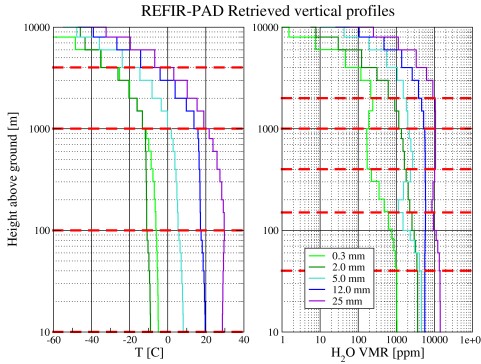

**Figure 15.** Vertical temperature and water vapor profiles obtained from the spectra in Figure 13, top panel, through the level 2 data analysis process. Red dashed lines show the selected fitting layers for temperature and water vapor.

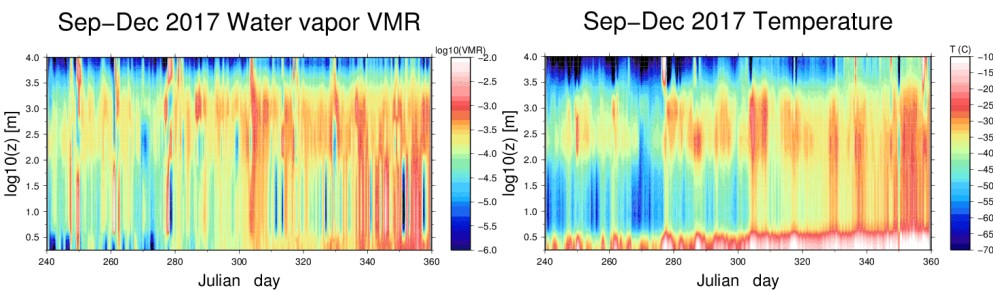

**Figure 16.** Vertical temperature and water vapor profile maps covering the Sepember – December, 2017 period. The warmer layer in the first meters corresponds to the optical path inside of the shelter and of the measurement chimney.

In order to better reflect the atmospheric modeling performed by LBLRTM the fitted profiles are shown as histograms following the layering structure adopted in the forward model. The logarithmic scale adopted for the representation of the vertical profile reflects the logarithmic spacing used in the layering, which derives from the decrease of vertical resolution with height that is inherent in the zenith-looking vertical sounding geometry.

5     A better visualization of the products of the retrieval process can be obtained by plotting the profiles vs. time as color maps. In Figure 16 are shown the temperature and water vapor maps obtained for the September – December period in 2017. Note that due to the installation of the instrument inside of an heated shelter, with a 2 m chimney connecting it to the outside air, the first meters of the retrieved profiles correspond to the shelter inside air.

    The profile maps presented in Figure 16 show the transition to Antarctic summer, with the onset of a diurnal cycle for the 10  temperature inversion, cycle that can be correctly resolved and characterized with the 12 minutes repetition rate of REFIR-PAD measurements.

The PWV is also provided as a level 2 data product. The accuracy in the determination of the PWV depends on the atmospheric conditions (total amount of water, presence of clouds) and ranges from 10-20% in the extremely dry conditions found in Antarctica to about 5% in mid-latitude atmosphere. Accuracy on the total PWV has been estimated through the error on water vapor column fitting, and validated with a microwave radiometer (Fiorucci, 2008; Bianchini, 2011).

Columnar amount of other tropospheric minor species with spectral lines in the REFIR-PAD measurement range can also be retrieved. For example nitrous oxide is obtained by adding an extra fit parameter which rescales the vertical $N_2O$ profile in the temperature and water vapor fitting process, making use of the 589 $cm^1$ spectral band. In figure 17, bottom panel, a time series of the retrieved $N_2O$ obtained from measurements performed in the September, 2017 – April, 2018 period is shown.

It should be noted that a similar approach could be also applied to methane, provided a suitable spectral window containing methane absorption features is added to the retrieval range. However, the main methane absorption feature overlaps with the absorption bands of the Mylar beam splitter substrate, so a different beam splitter design (e. g. based on Polypropylene) would be needed for an efficient methane total column retrieval.

A different consideration must be made for ozone column retrieval: while a strong ozone emission band is present in the REFIR-PAD operating spectral interval, most of the ozone lies in the stratosphere where the temperature retrieval, mainly relying on the Carbon Dioxide $\nu_2$ band, has no sensitivity. Thus to correctly interpret the emitted radiance due to the ozone band, stratospheric temperatures must be provided as an external input. This can be done through radiosounding or an auxiliary sensor as a stratospheric Raman LIDAR (Bianchini, 2014).

The ozone retrieval process makes use of the 920-1070 $cm^{-1}$ spectral range. The retrieval grid, obtained through Jacobian analysis as in case of temperature and water vapor retrieval, features 3 fitted levels in the 12-24 km altitude range.

In Figure 17, top panel, are shown ozone columnar amounts obtained in the September 2017 – April 2018 period, together with the available NOAA OMI/OMPS ozone time series data[1], calculated for the ground pixel corresponding to Concordia station.

While a noticeable offset in ozone data is present and needs to be investigated further, the temporal variability is in good agreement with the satellite data, and the vertical variability observed in the retrieved 3-points profile shows a good correlation with the rapid variations in the columnar amounts. This can be explained with the fact that Dome C lays on the edge of the polar vortex region, so that it can enter and exit the vortex region depending on atmospheric transport.

## 11 Level 2 auxiliary outputs

The instrumental parameters obtained from the level 2 data analysis process provides a valuable tool to characterize the quality of the measured spectra and the performance of the instrument. Figure 18 shows the result of the analysis of the instrumental parameters in the 2016-2017 period. During these two years the REFIR-PAD instrument operated continuously without any significant maintenance.

---

[1] http://www.esrl.noaa.gov/gmd/grad/neubrew/SatO3DataTimeSeries.jsp

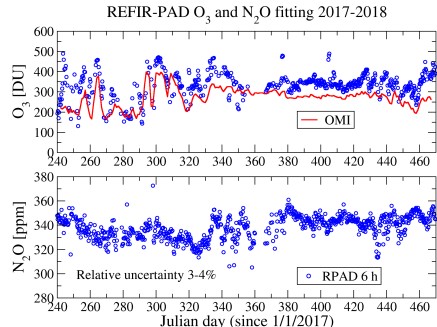

**Figure 17.** Time series of Ozone and Nitrous Oxide columnar values retrieved from The REFIR-PAD measurements acquired in the September 2017 – April 2018 period. For reference, corresponding OMI measurements over the Dome C region pixel are also shown.

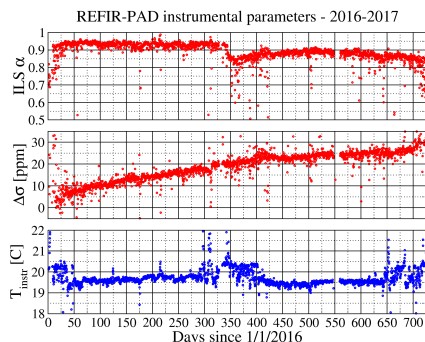

**Figure 18.** REFIR-PAD instrumental parameters for the 2016-2017 period. Top panel: ILS coefficient $\alpha$. Center panel: laser frequency error $\Delta\sigma$. Bottom panel: Instrument temperature.

In December 2015 the laser source was replaced due to a malfunction inducing sudden mode jumps, and a laser frequency calibration was performed. Long term frequency stability of the laser source on the following two years can be evaluated from the laser frequency error $\Delta\sigma$, shown in Figure 18, center panel.

Laser frequency appears to be subjected to a slow drift that accumulated a total deviation of about 25-30 ppm since the initial calibration.

This behaviour derives mostly from laser diode aging, since it is not correlated with the instrument temperature (shown in Figure 18, bottom panel) as, instead, would be the effects due to the control electronics. The observed drift is also about an order of magnitude larger than the temperature drifts estimated using the calculated thermal coefficients (see Section 7).

The ILS coefficient $\alpha$ (top panel) also gives useful insights on the instrument performances in the selected period: it can be seen that during the summer season between 2016 and 2017 (the center part of the plot), larger than usual temperature

fluctuation are present. This is due to the higher outside temperatures and also due to personnel working inside of the shelters, which is much more frequent during summer.

The temperature fluctuations impact on the ILS coefficient due to thermally induced optical misalignment, which is only partially recovered in the following winter season, so that a yearly optics check and realignment to be performed in summer is
desirable even if not mandatory.

## 12   Conclusions

The REFIR-PAD spectroradiometer has proved to be a reliable and versatile tool for the remote sensing of the radiative properties, composition and thermal structure of the troposphere.

The instrument is capable of providing a wealth of information with a measurement repetition rate of the order of 10 minutes,
fast enough to resolve all the relevant cloud-free atmosphere processes, which is the reference case in this work.

The currently available data products are:

- Atmospheric emitted radiance spectra in the 100-1500 $cm^{-1}$ range with 0.4 $cm^{-1}$ resolution and 0.85 mW/(m$^2$sr cm$^{-1}$) accuracy.

- Tropospheric water vapor and temperature vertical profiles with up to 5 independently fitted points.

- Total columnar precipitable water vapor (PWV) with an accuracy ranging from 5% to 20% depending on the total humidity and atmospheric conditions.

- Columnar amounts of minor species as nitrous oxide and ozone.

- Cloud optical thickness in the atmospheric transparency window region (800-1200 $cm^{-1}$).

The instrument operates at room temperature, is fully autonomous and allows for complete remote control of the configura-
tion parameters, thus is perfectly suitable for operation in remote and extreme environment, as demonstrated by more than 6 years of continuous operation in the Antarctic station Concordia.

It should be noted that this specific location provides itself an unique dataset, since no similar instruments are operating continuously in the Antarctic continent. Currently the REFIR-PAD instrument is operated in the framework of two different projects funded by the Italian Antarctic Program in the perspective of reaching at least a decade-long measurement time series.
Future outlooks include the development and test of new beam splitter designs to overcome the spectral band limitations posed by the use of a Mylar substrate. The development of Polypropylene-based beam splitters is currently in progress, and will allow to fully exploit the 1100-1400 $cm^{-1}$ spectral region to add new products (e.g. methane) to the currently available ones.

*Acknowledgements.* We would like to acknowledge the Italian Antarctic Program, Programma Nazionale di Ricerca in Antartide (PNRA) for the funding for the following research programs, that have allowed to perform REFIR-PAD measurements since December 2011 at Concordia Station, Antarctica: project PRANA (Proprietà Radiative del vapore Acqueo e delle Nubi in Antartide) 2009/A04.03 2011-2013, project COMPASS (COncordia Multi-Process Atmospheric StudieS) 2013/AC3.01 2013-2016, and the currently active projects DOCTOR

5   (DOme C Tropospheric ObserveR) 2016/AC3.02 and FIRCLOUDS (Far Infrared Radiative Closure Experiment For Antarctic Clouds) 2016/AC3.03.

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
