# Peer review of "A Fourier transform spectroradiometer for ground-based remote sensing of the atmospheric downwelling long-wave radiance"

_Atmospheric Measurement Techniques, 2018_

## Referee Comment (RC1) · Anonymous Referee #1 · 13 Aug 2018

Comments to the authors: I believe that this paper was written to serve as the "instrument paper" for the REFIR-PAD; i.e., the source of all of the technical details needed to understand the instrument, the corrections that are applied to account for instrument artifacts, how it is calibrated, and the uncertainties in its radiance observations. Furthermore, it also provides some examples of level-2 products that can be derived from these observations.

From this perspective, I was expecting a paper to have a lot of details associated with the instrument, especially in sections 2-5. I was also expecting the information to be presented in a similar manner to other interferometer instrument papers (e.g., the

ARIES paper by Wilson et al in JTECH 1999, the AERI papers by Knuteson et al. in JTECH 2004); in other words, a careful presentation of all of the details so that I understand the instrument, how it is radiometrically and spectrally calibrated, its uncertainties, its operating characteristics, etc. This paper on the REFIR-PAD did provide some of the information, but there was still a lot missing. I will detail some of these areas below.

Generally speaking, more references are needed in this paper. Furthermore, the majority of the references (28 out of 34) were by the authors of this paper; are there no other papers written by outsiders that are relevant to this study?

Page1, Line 40: This is true only in clear sky horizontally homogeneous scenes. This approach will generally not work when there are clouds overhead

Page 3, line 10: "that result critical in the delicate process" is very awkward. Please rephrase

Page 3, line 75: Is the spectral calibration procedure similar to that in Knuteson et al. JTECH 2004? What is the spectral region used for this calibration?

Page 3, line 85: What are the details of this scene mirror? Is it gold plated? What polarization properties?

Page 3, line 85: What are the properties of these blackbodies? Emissivity spectra, operating temperatures, etc. How stable are they? What is the shape, arrangement of the thermistors, gradients, etc?

Page 5, line 68: How high? What is the IWV amount? What does a LBL radiative transfer model suggest the radiance should be for this condition? Is the small bias shown in the figure due to the small amount of atmospheric emission (which could be confirmed by a RT model), or is it a real instrument artifact?

Page 6, line 4 and elsewhere in that paragraph: Should be mW / (m**2 sr cm**-1).

Page 6, line 58: What is this chain? Does it use the Revercomb technique to calibrate in complex radiance? How is non-unity emissivity of the BBs handled?

Page 7, line 4: what is "assimilable"? Perhaps you mean "similar" ?

Figure 8 and in the text: Is the imaginary component of the calibrated spectra zero with some noise? Would be good to see that, esp since fig 8 shows some unbalanced spectra with significant phase signals.

What is a typical noise spectrum for a standard radiance measurement?

Page 7, line 28: Are these instrumental parameters (line shape, spectral calibration) not stable with time ? If that is true, why is this so?

Page 7, line 32: How were these number of layers determined; e.g., why a 4pt temperature profile? Turner and Löhnert JAMC 2014 using mid-infrared portion of the spectrum suggest that there is ∼6 pieces of information on temperature (and similar for water vapor when the IWV is small), so I would have assumed that the REFIR-PAD observations would have had at least this number of pieces, unless the noise level is much larger than the AERI used in the T/L paper (which is why the noise spectrum needs to be shown).

Page 7, line 33: "tipical" is misspelled.

Page 7, line 34: Is the entire spectral range of the REFIR-PAD observations used in the retrieval?

Fig 11: The spectral structure of the radiance observations in the 15 $\mu$m band suggest that there is an inversion in the purple spectrum, and that the lapse rate is markedly different for the dark green vs. light green profiles. But these characteristics don't show up in these retrieved profiles shown in Fig 11 (or at least are not obvious to my eye). Is this due to the low number of vertical layers?

Page 8, line 6: How were these accuracies determined? Are they just the uncertainties

from the propagation through the retrieval, or comparison against other obs?

If you are going to talk about the retrievals in this paper, then more information needs to be provided so that the reader does not have to search through all of the references to get this information. Please include a discussion on the basic retrieval framework, what assumptions are made, the forward model used, any prior data used to constrain the solution, etc.

Page 8, line 29: I don't think that an interferometer like REFIR-PAD can be considered a "relatively simple tool". Even compared with other spectroradiometers this instrument is pretty advanced. Now, perhaps is operating characteristics make it easy to deploy and it can run autonomously, and that is what the authors are referring to here. If so, then there is little information in this paper about the long-term calibration stability and responsivity of the instrument, other than the oblique reference that some instrument parameters need to be retrieved (see above)...

Page 8, line 34: this instrument cannot "resolve all relevant atmospheric processes". For example, 10-minute resolution is not able to resolve the rapid changes in cloud optical properties as they advect over the sky port of this instrument.

Page 8, line 43: The o3, ch4, and n2o retrievals were demonstrated here, and references to papers that show this are few / none.

Page 8, line 47: As indicated above, you haven't spoken about the long-term operations at all, and certainly not the ability to remotely control the instrument (this is the first mention of it). What are the "relevant settings" ?

Page 8, line 55: "aknowledge" is misspelled.
* * *

---

## Referee Comment (RC2) · Anonymous Referee #2 · 14 Aug 2018

**General comments**

This paper describes the optical setup, some aspects of the performances and exemplary data products of the ground based REFIR-PAD far infrared FTS instrument. During the last 15 years the instrument has been deployed on various missions ranging from a stratospheric balloon flight, installations in alpine sites to the current installation in the CONCORDIA Antarctic station. The principal technical design, a comprehensive performance analysis and results of previous missions have already been published elsewhere. The present article gives an overview of the current status of the instrument. Some performance issues like instrumental line shape and radiometric offset are rediscussed without highlighting the added value to the already

published analysis (Bianchini, 2008b). The paper does not provide and/or explain a new scientific data set.

The theme of the article fits well within the scope of AMT and the article is clearly written, but the current scope does not provide enough new information as compared to already existing literature. No substantial new concept or data is presented. Therefore I recommend to publish the article only after major extension/revision. In particular the performance analysis part has to clearly state the new insights gained in comparison to earlier papers and the data section needs to present a more comprehensive overview of the Antarctic dataset.

**Specific comments**

Section 3 discusses instrument lineshape. Please highlight the new insights gained relative to the information provided earlier (Fig. 4 of the current paper and Fig. 17 of the Bianchini, 2008b paper seem to be identical).

Section 4 discusses detectors and data acquistion electronics. What is different to the analysis performed in section 2.1 of the abovementioned paper (Figure 6 of the current paper and fig. 3 and 5 of the 2008 paper seem to convey the same information)?

Section 5 discusses radiometric performances. A statistical analysis of offset values in one atmospheric window region is presented. Again the value of this analysis in the context of the existing radiometric performance analysis needs to be stated more clearly. Instrument offset will be wavenumber dependent. An analysis of the radiometric offset in other spectral regions is of interest.

Section 6 states to discuss spectroscopic performances. It then describes qualitatively the agreement between an analytical instrument model and laboratory measurements. The model is not detailed and there is no quantitative discussion of the discrepancies between model and measurement. No attempt is made to derive figures of merit and compare them with requirements. The title of the section is misleading and the description of model and results is not sufficing to provide insights. The section should either be omitted or renamed and significantly extended.

[Figure]

Section 7, last sentence: I do not think that one offset measurement at 835 cm-1 is sufficient to derive a relative uncertainty for the whole wavenumber region from 200-667 cm$-1$. The deduction could possibly be made with the help of the instrument model, but then this needs to be demonstrated.

Section 8 shows an exemplary L2 data set. Yet no comprehensive data set is presented (e.g. a time series of measurements), no scientific interpretation is provided and no quality assessment (e.g. validation through other data) is made. There is no supplement with data or information about where the data could be accessed. It is mentioned in section 8 / L2 products that the retrieval of methane requires a hardware modification of the instrument. In the conclusions section, Methane is mentioned as provided data product, though.

Section 9 (conclusions) reiterates the properties of the instrument without providing real conclusions or an outlook.

---

## Author Comment (AC1) · 19 Oct 2018

First of all I would like to thank the reviewer for the thoughtful and in-depth review, which had provided good tools for improving this work.

I'll answer, as much as possible, to each of the single questions posed:

*Generally speaking, more references are needed in this paper. Furthermore, the majority of the references (28 out of 34) were by the authors of this paper; are there no other papers written by outsiders that are relevant to this study?*

Yes, definitely, I must say that the choice of the references has been biased by the

attempt of providing information about the specific instrument described in the paper without repeating what was already published, while the use of references to show other relevant works in the field has been quite overlooked. The introductory section has been reorganized in order to add references to previous works that are relevant to the paper topic.

*Page1, Line 40: This is true only in clear sky horizontally homogeneous scenes. This approach will generally not work when there are clouds overhead*

True, it will be clarified that this consideration, so as other to which a similar remark could apply, are referred to clear sky conditions, since the considerations and problems related to the study of clouds are out of the scope of this work.

*Page 3, line 10: "that result critical in the delicate process" is very awkward. Please rephrase*

Rephrased in the revision.

*Page 3, line 75: Is the spectral calibration procedure similar to that in Knuteson et al. JTECH 2004? What is the spectral region used for this calibration?*

The procedure has some similarities, but is not the same: in order to provide a robust algorithm that can operate in all the possible measurement conditions, it has been chosen to use the hot blackbody acquisitions for frequency calibration. This will not to perform an independent frequency calibration of each spectrum, but has the advantage of using the much more reproducible absorption spectrum due to $CO_2$ on the about 1.5 m optical path inside of the instrument. The calibration operates in two phases, first a rough peak finding algorithm detects the Q band center, then the whole P band in the 635-665 cm$^{-1}$ spectral region is fitted using the simplified $0.9 \cdot \text{sinc} + 0.1 \cdot \text{sinc}^2$ lineshape (see attached Figure 1).

This two-step process is required due to the fact that the diode laser could in principle have a frequency shift larger than the spacing of the $CO_2$ lines and this could induce a

systematic error due to a "skip" of one or more lines. The description of the algorithm, and all the information and figures here provided will be added to the revised paper.

*Page 3, line 85: What are the details of this scene mirror? Is it gold plated? What polarization properties?*

The folding mirror is bare gold on an aluminum substrate, and has been characterized through laboratory measurement in order to provide the small correction needed for the calibration (correction that is applied using the monitored mirror temperature).

The effect of polarization is estimated as negligible, taking into account the fact that the instrument is not operating in polarization mode and the zenith scene, in clear sky condition (the operating conditions taken as a reference in this paper) is not polarized.

*Page 3, line 85: What are the properties of these blackbodies? Emissivity spectra, operating temperatures, etc. How stable are they? What is the shape, arrangement of the thermistors, gradients, etc?*

Detailed information on the blackbody sources is available in [Palchetti et al., Infrared Physics Technology 51 (2008)], but in order to improve text readability the main details on the blackbody performances will be added to the text. Specifically, the emissivity is better than 0.999 and the operational temperatures are between 10 and 80 °C. Stability is about same order of the temperature reading uncertainty (0.3 K), while gradients are within 0.5 K.

It should be noted that the calibration procedure compensates for linear temperature drifts of the blackbody temperature (specifically, of the reference blackbody source placed on the second input, which is providing a common reference to all the acquisitions).

*Page 5, line 68: How high? What is the IWV amount? What does a LBL radiative transfer model suggest the radiance should be for this condition? Is the small bias shown in the figure due to the small amount of atmospheric emission (which could be*

*confirmed by a RT model), or is it a real instrument artifact?*

The plot was obtained exploiting the full RHUBC-II dataset, acquired from the Cerro Toco site at about 5500 m a.s.l. in the Atacama region. The dataset nevertheless included some acquisitions characterized by 1 mm or more in terms of PWV. This in fact hasn't been a good choice, since the atmospheric residual emission is the main cause of the small offset observed: an improved version of the figure (Figure 3 here attached) has been made using only measurement selected to have a PWV < 0.6 mm, which would give an offset negligible with respect to the instrument estimated accuracy.

*Page 6, line 4 and elsewhere in that paragraph: Should be mW / (m\*\*2 sr cm\*\*-1).*

Corrected in the revision.

*Page 6, line 58: What is this chain? Does it use the Revercomb technique to calibrate in complex radiance? How is non-unity emissivity of the BBs handled?*

The level 1 data analysis is described in detail in [Bianchini et al., ACP 8 (2008)], it makes use of the complex calibration [Revercomb et al., Appl. Opt. 27 (1988)], while for the blackbody sources a specific mathematical model has been developed [Palchetti et al., J. Infr. Phys. Tech. 51 (2008)], it will be made an effort to add as much information as possible from the above mentioned references in order to make the paper more readable.

*Page 7, line 4: what is "assimilable"? Perhaps you mean "similar" ?*

Corrected in the revision in order to avoid the confusion with the most used meaning of "assimilation" in this field...

*Figure 8 and in the text: Is the imaginary component of the calibrated spectra zero with some noise? Would be good to see that, esp since fig 8 shows some unbalanced spectra with significant phase signals. What is a typical noise spectrum for a standard radiance measurement?*

A figure (Figure 4 here below) will be added, showing a typical calibrated radiance spectrum (obtained by the real part of the result of the calibration procedure), and the corresponding discarded imaginary part, which contains only the noise, confirming the reliability of the complex calibration procedure.

*Page 7, line 28: Are these instrumental parameters (line shape, spectral calibration) not stable with time ? If that is true, why is this so?*

An analysis of the long-term stability of the instrument calibration will be added, in which it is shown that, in absence of spurious effects, the laser stability allows for a $< 30$ ppm frequency calibration accuracy over a period of 2 years (dominated by a drift due to laser aging). The instrumental line shape is instead affected by misalignment that can occur in case of large thermal excursions of the instrument (Figure 5 here attached).

This analysis involves a 2-year long period in which no maintenance has been performed on the instrument. The observed effect around the middle of the considered period arises from operations performed on other instrumentation inside of the shelter where REFIR-PAD is installed, operations that caused some level of disturbance due to temperature fluctuations and vibrations.

*Page 7, line 32: How were these number of layers determined; e.g., why a 4pt temperature profile? Turner and Löhnert JAMC 2014 using mid-infrared portion of the spectrum suggest that there is 6 pieces of information on temperature (and similar for water vapor when the IWV is small), so I would have assumed that the REFIR-PAD observations would have had at least this number of pieces, unless the noise level is much larger than the AERI used in the T/L paper (which is why the noise spectrum needs to be shown).*

The number and altitudes of the fitted levels have been determined by an analysis of Jacobians and a subsequent tuning to avoid oscillations in absence of a-priori constraints, as described in [Bianchini et al., JGR 116 (2011)]. Please note that the noise

level on REFIR-PAD spectra is actually higher than that on AERI spectra, this is mainly due to the use of uncooled pyroelectric detectors instead of cooled MCT/InSb.

*Page 7, line 33: "tipical" is misspelled.*

Corrected in revision.

*Page 7, line 34: Is the entire spectral range of the REFIR-PAD observations used in the retrieval?*

Only a subset of the REFIR-PAD spectra between 350 and 850 $cm^{-1}$ is used for the temperature and water vapor profile retrieval. This in general provides consistent results in a wide range of atmospheric conditions. A different subset, between 920 and 1070 $^{-1}$ is used for the ozone column retrievals. This will be clarified in the revised text.

*Fig 11: The spectral structure of the radiance observations in the 15 m band suggest that there is an inversion in the purple spectrum, and that the lapse rate is markedly different for the dark green vs. light green profiles. But these characteristics don't show up in these retrieved profiles shown in Fig 11 (or at least are not obvious to my eye). Is this due to the low number of vertical layers?*

As can be seen in the attached plots (Figures 6-9 here below), showing the fitted spectra in the $\nu_2$ band region, the fitting residuals are well inside of the measurement error, apart for one case in which I noticed that a laser mode jump happened, distorting the averaged spectrum (Figure 6 in this document, corresponding to the light green spectrum in the paper). In this case I removed the spectra showing mode jumps from the average, and as a consequence now the chi-square is better (Figure 7 here, light green spectrum after correction). Anyway, there isn't a relevant change in the resulting temperature profiles, and no significant inversion is present.

Most of the effect seen on the $\nu_2$ band can be attributed to the $CO_2$ present inside and nearby the instrument enclosure but outside of the calibration path so it doesn't cancel out. This doesn't give significant effects on the retrieval, since the overall contribution

to the chi square is negligible. In Figure 8 here below (corresponding to the purple spectrum in the paper) the instrument was slightly cooler than the environment, being placed in cool shade on a warm spring day, while in Figure 9 here below (corresponding to the dark green spectrum in the paper) it's the opposite since the instrument was operating outside in February. Please note that I made these plots for example purposes, In my opinion these shouldn't be included in the revised text, unless it is deemed absolutely necessary.

*Page 8, line 6: How were these accuracies determined? Are they just the uncertainties from the propagation through the retrieval, or comparison against other obs? If you are going to talk about the retrievals in this paper, then more information needs to be provided so that the reader does not have to search through all of the references to get this information. Please include a discussion on the basic retrieval framework, what assumptions are made, the forward model used, any prior data used to constrain the solution, etc.*

Accuracy on the total PWV has been estimated through the error on water vapor column fitting, and validated with a microwave radiometer [Fiorucci et al., JGR 113 (2008)], [Bianchini et al., JGR 116 (2011)]. I understand the fact that repeating some information that is in the cited references could greatly improve readability, but I have been advised (by the other reviewer) not to introduce information that is redundant with other published papers. An effort will be made anyway to add the required information in the revised text.

The retrieval was performed by using the MINUIT routine which is part of the CERNlibs. The subroutine MIGRAD, based on the Davidon-Fletcher-Powell (DFP) algorithm, was used to minimize the chi square cost function given by:

$$\chi^2 = (y - F(X))^T S_y^{-1}(y - F(x)) \tag{1}$$

where $y$ and $x$ are the vector of the measurements and the state of the atmosphere

respectively, F is the forward model (LBLRTM version 12.2 in our case) and $S_y$ is the diagonal VCM for the measurements. The DFP algorithm, on which the MINUIT MIGRAD routine is based, is a quasi-Newton method which does not require the calculation of the jacobians at each iteration but uses an approximated form. This algorithm updates the inverse hessian matrix calculating the derivatives just at the first step and then using the iterative formula shown above. The same fitting approach which was applied in a previous works [Bianchini et al., JGR 116 (2011)], was used in this paper. No a-priori information was assumed as regularly done in a Bayesian approach, such as optimal estimation, and the initial guess is represented by a local monthly climatology, obtained averaging over a set of radiosoundings daily performed at Dome-C. Since no a-priori information was used to constrain the solution and no regularization was introduced, to avoid the oscillation effects due to the ill-conditioning of the problem this approach requires to limit the number of retrieved parameters, hence the number of fitted levels both for water vapor and temperature profiles is equal to the number of degrees of freedom (DOF). The DOF were derived from a preliminary study performed through singular values decomposition of the Hessian matrix which includes Jacobian and the measurements noise.

*Page 8, line 29: I don't think that an interferometer like REFIR-PAD can be considered a "relatively simple tool". Even compared with other spectroradiometers this instrument is pretty advanced. Now, perhaps is operating characteristics make it easy to deploy and it can run autonomously, and that is what the authors are referring to here. If so, then there is little information in this paper about the long-term calibration stability and responsivity of the instrument, other than the oblique reference that some instrument parameters need to be retrieved (see above)*

I agree that the choice of the term "simple" is at least misleading, if not plainly wrong, the sentence will have to be rephrased in order to stress the simplicity of operation and the ruggedness that allow for minimal need for interventions and maintenance. As stated before, long-term stability of instrument parameters will be described in the

revised text, so will be the hardware that allow for remote operation and management.

*Page 8, line 34: this instrument cannot "resolve all relevant atmospheric processes". For example, 10-minute resolution is not able to resolve the rapid changes in cloud optical properties as they advect over the sky port of this instrument.*

Yes, as stated before, and as will be explicitly described in the revised text, this has to be intended as in clear sky conditions.

*Page 8, line 43: The o3, ch4, and n2o retrievals were demonstrated here, and references to papers that show this are few / none.*

More details on the procedure to retrieve $N_2O$ and $O_3$ will be added: in the first case it is an extra parameter that rescales the vertical $N_2O$ profile in the T/WV fitting process, making use of the 589 cm$^{-1}$ spectral band. In the case of $O_3$ a separate fitting process is used, operating in the 920-1070 cm$^{-1}$ spectral region with a total of three fitted levels. In Figure 10 here attached are shown some results obtained in the September 2017 – April 2018 period, in case of ozone the available OMI data are also shown for comparison.

While a noticeable offset in ozone data is present and needs to be investigated further, the temporal variability is in good agreement with the satellite data, and the vertical variability observed in the retrieved 3-points profile shows a noticeable variation in the vertical ozone structure in coincidence with the rapid variations in the columnar amounts (Figure 11 here below). This can be explained with the fact that Dome C is on the edge of the polar vortex region.

*Page 8, line 47: As indicated above, you haven't spoken about the long-term operations at all, and certainly not the ability to remotely control the instrument (this is the first mention of it). What are the "relevant settings" ?*

A section of the revised text will describe the infrastructure allowing for remote control and management (remote shell connection, transfer of selected and preprocessed

data through low-bandwidth connection, auxiliary instrumentation and subsystems as thermal stabilizers)

*Page 8, line 55: "aknowledge" is misspelled.*

Corrected in the revised text.

[Figure]

REFIR-PAD, frequency calibration CO$_2$ ν2 band fitting

Normalized uncalibrated spectrum [a. u.]

—— measured
—— fitted
—— residuals

Wavenumber [cm$^{-1}$]

**Fig. 1.**

[Figure]

Fig. 2.

[Figure]

REFIR-PAD - average radiance (828-839 cm⁻¹ interval)

Gaussian fit ($x_0 = 6.10^{-6}$, $\sigma = 7.10^{-4}$)

Samples No.

Radiance [W/ m$^2$ sr cm$^{-1}$]

**Fig. 3.**

REFIR-PAD Dome C 1/3/2018, sequence #3, channel 1,

*(figure: Spectral radiance [W/m² sr cm⁻¹] vs Wavenumber [cm⁻¹]; legend: real part (red), imaginary part (blue), uncertainty (black))*

**Fig. 4.**

**REFIR-PAD instrumental parameters - 2016-2017**

[Figure: three stacked scatter plots versus "Days since 1/1/2016". Top panel: ILS α (red), y-axis from 0.5 to 1. Middle panel: Δσ [ppm] (red), y-axis from 0 to 30. Bottom panel: T_instr [C] (blue), y-axis from 18 to 22. X-axis from 0 to 700.]

**Fig. 5.**

REFIR-PAD, Cerro Toco 22/10/2009, clear sky, 0.28 mm PWV, $\chi^2$=6.1

Legend:
- measured
- fitted
- residuals

**Fig. 6.**

REFIR-PAD, Cerro Toco 22/10/2009, clear sky, 0.32 mm PWV, $\chi^2$=1.3

*Figure plot with axes: Radiance [W/m$^2$ sr cm$^{-1}$] vs Wavenumber [cm$^{-1}$], legend: measured, fitted, residuals*

**Fig. 7.**

REFIR-PAD, Sesto Fiorentino 26/05/2011, clear sky, 25.5 mm PWV, $\chi^2$=0.92

**Fig. 8.**

REFIR-PAD, Sesto Fiorentino 25/02/2011, clear sky, 5.1 mm PWV, $\chi^2$=0.44

**measured**
**fitted**
**residuals**

Radiance [W/m$^2$ sr cm$^{-1}$]

Wavenumber [cm$^{-1}$]

**Fig. 9.**

REFIR-PAD O$_3$ and N$_2$O fitting 2017-2018

Fig. 10.

[Figure]

**Fig. 11.**

---

## Author Comment (AC2) · 19 Oct 2018

Before proceeding, thanks for the time spent in your review, I'll try to answer to all the questions adding the requested information whenever possible, information that will be also included in the reviewed text.

*The theme of the article fits well within the scope of AMT and the article is clearly written, but the current scope does not provide enough new information as compared to already existing literature. No substantial new concept or data is presented. Therefore I recommend to publish the article only after major extension/revision. In particular the performance analysis part has to clearly state the new insights gained in comparison to*

*earlier papers and the data section needs to present a more comprehensive overview of the Antarctic dataset.*

Section 7 has been reorganized and more information has been provided to describe the available dataset and the Antarctic campaign.

*Section 3 discusses instrument lineshape. Please highlight the new insights gained relative to the information provided earlier (Fig. 4 of the current paper and Fig. 17 of the Bianchini, 2008b paper seem to be identical).*

Yes, actually the instrumental lineshape in the far-infrared region is expected to be quite ideal, by design, so the figure has been changed (see Figure 1 here below) showing the ILS at two different wavenumbers, both in the far-infrared and at the upper limit of the operating spectral band, for two different spectral sampling values (0.5 and 0.25 cm$^{-1}$). This will integrate the data presented in Figure 5 in the paper, that shows the retrieved lineshape coefficient at different wavenumbers, providing a more complete characterization of the instrumental line shape as a function of spectral sampling and wavenumber.

*Section 4 discusses detectors and data acquistion electronics. What is different to the analysis performed in section 2.1 of the abovementioned paper (Figure 6 of the current paper and fig. 3 and 5 of the 2008 paper seem to convey the same information)?*

Also in this case, as expected, the instrumental parameters have not changed significantly in time. A choice was made to present some data that is redundant with the published papers: as it has been noted by the other reviewer, in some cases information should be repeated in order to allow readability without having necessarily to keep at hand also all the references. Anyway, the bottom panel of Figure 6 (in the paper) is redundant and will be omitted, while the top panel which corresponds to the last performed characterization involving the current detectors will be kept since it contains new information.

*Section 5 discusses radiometric performances. A statistical analysis of offset values in one atmospheric window region is presented. Again the value of this analysis in the context of the existing radiometric performance analysis needs to be stated more clearly. Instrument offset will be wavenumber dependent. An analysis of the radiometric offset in other spectral regions is of interest.*

Yes, the observation is correct, but we do not have a similar check that can be performed over the whole spectral range and continuously during a multi-year deployment of the instrument. Thus this kind of estimate, even if related to a narrow spectral range, is relevant. An estimate covering the full spectral range can be obtained through an external reference blackbody placed on the instrument measurement port, but this requires dedicated measurements and cannot be performed during remote operation.

Figure 2 here below shows the results of such a calibration measurement. Even in this case the calibration accuracy is quite constant over the relevant spectral range and well inside of the measurement uncertainties. Below 300 cm$^{-1}$ and in the correspondence of the beam splitter substrate absorption bands the measurement errors are prevalent and it is difficult to quantify the actual calibration accuracy.

*Section 6 states to discuss spectroscopic performances. It then describes qualitatively the agreement between an analytical instrument model and laboratory measurements. The model is not detailed and there is no quantitative discussion of the discrepancies between model and measurement. No attempt is made to derive figures of merit and compare them with requirements. The title of the section is misleading and the description of model and results is not sufficing to provide insights. The section should either be omitted or renamed and significantly extended.*

I agree that the title "Spectroscopic performances" is without doubt a misnomer, this is my fault because I left the title as it was through various changes of the text, it will be changed in "Instrument mathematical modeling". The scope of the model is to provide a qualitative analysis devoted to the understanding and, if possible, correction,

of the various instrumental effects. The main scope is to understand the effect of non-ideal beamsplitters and to devise the best way to obtain optical path difference compensation, i.e. minimizing the phase variations across the spectral range.

This is achieved mainly with the correct orientation of the beam splitters, but also small layer thickness differences can give a measurable effect. All this will be better explained in the revised text, along with a more in-depth description of the model used and the corrected title.

*Section 7, last sentence: I do not think that one offset measurement at 835 cm-1 is sufficient to derive a relative uncertainty for the whole wavenumber region from 200-667 cmôĂĂĂ1. The deduction could possibly be made with the help of the instrument model, but then this needs to be demonstrated.*

A new figure (number 2 in this document) showing a characterization of the radiometric uncertainty over a wider spectral range will be added to the revised paper. Nevertheless, the results shown in Figure 7 in the paper (which has been updated with a better selection of the analyzed measurements, Figure 3 here below) have their specific purpose in providing a way to check the consistency and stability of the calibration without the need of dedicated measurements. This is of particular importance in case of a remote deployment of the instrument. The sentence will be rephrased to clarify this.

*Section 8 shows an exemplary L2 data set. Yet no comprehensive data set is presented (e.g. a time series of measurements), no scientific interpretation is provided and no quality assessment (e.g. validation through other data) is made. There is no supplement with data or information about where the data could be accessed. It is mentioned in section 8 / L2 products that the retrieval of methane requires a hardware modification of the instrument. In the conclusions section, Methane is mentioned as provided data product, though.*

For what concerns data validation, this has been performed during previous campaigns, see [Fiorucci et al. JGR 113 (2008)], [Bianchini et al. JGR 116 (2008)], [Turner

et al. GRL 39 (2012)] in which REFIR-PAD measurements and retrievals have been compared with other instruments, both FTS and microwave radiometers.

The presented level 2 data examples were meant more as a way to show the capability of the instrument/data analysis process to operate across a wide range of atmospheric conditions (in terms of temperature and water vapor content) rather than showing a long-term time series.

While a complete discussion of a long term time series of data would need a complete scientific discussion on its own (and it's possibly out of the scope of this "instrument" paper), however a data set covering a period of about an year will be provided, showing all the level 2 products that are available at the moment (temperature and water vapor profiles, $N_2O$ and $O_3$ columns, see Figures 4, 5 and 6 here below).

Please note that due to research project constraints, the actual data set cannot be released at the moment, since there is a clause that restricts the access to project participants for the whole project's duration and the following two years.

Unfortunately, for what concerns methane, no data can be provided since the Polypropylene-based beamsplitters even if realized, have not still been used on the field. For what concerns nitrous oxide, it is a secondary output of the temperature and water vapor profile retrieval, it makes use of the 595 cm$^{-1}$ band and just rescales the initial guess profile (constant in value throughout the troposphere). The values shown in Figure 6 here below are expressed in ppm of VMR at ground due to the nature of the rescaled profile. Ozone fitting is performed separately using the 920-1070 cm$^{-1}$ spectral range and a 3-point vertical profile, the results shown in Figure 6 are compared with the available OMI data for the Dome C site.

*Section 9 (conclusions) reiterates the properties of the instrument without providing real conclusions or an outlook.*

The conclusions section will be rewritten in order to cite explicitly the importance and

uniqueness of the long-term dataset that has been and is still being acquired by the REFIR-PAD instrument. Outlooks for future development of the instrument (like the use of Polypropylene based beamsplitter and the new possibilities opened by this) and for future uses of the acquired dataset will also be presented.

———————————————

[Figure]

**REFIR-PAD Instrumental Line Shape Analysis**

Spectral sampling: top row (red) 0.25 cm$^{-1}$, bottom row (blue) 0.5 cm$^{-1}$

Plot grid of four panels. Y-axis: Amplitude [a. u.], X-axis (top): Wavenumber - 526 [cm$^{-1}$], X-axis (bottom right): Wavenumber - 1430 [cm$^{-1}$]. Legends: $\alpha = 0.9$, $\alpha = 0.3$, $\alpha = 1.0$, $\alpha = 0.8$.

**Fig. 1.**

BBext-comp.agr

Figure showing two stacked plots. Top plot: Rad vs Wavenumber with Measurement (red) and BB emission (black). Bottom plot: BT [K] vs Wavenumber with Difference (grey), BB uncertainty (blue), Meas. uncertainty (black).

**Fig. 2.**

REFIR-PAD - average radiance (828-839 cm$^{-1}$ interval)

Gaussian fit ($x_0$=6.10$^{-6}$, $\sigma$=7.10$^{-4}$)

*Samples No.*

Radiance [W/ m$^2$ sr cm$^{-1}$]

**Fig. 3.**

[Figure]

[Figure]

[Figure]

**Fig. 4.**

[Figure]

**Fig. 5.**

**REFIR-PAD O$_3$ and N$_2$O fitting 2017-2018**

O$_3$ [DU]

— OMI

N$_2$O [ppm]

Relative uncertainty 3-4%

o   RPAD 6 h

Julian day (since 1/1/2017)

**Fig. 6.**

---

## Author Response (AR2)

Dear Editor,

Here below is the list of the required corrections with my comments, point by point. All the modifications have been applied as requested, the only exception being the captions to figures, which are referring to the 2-column format, where multiple panels appear one on top of the other and not side by side as in the discussion format.

Thanks again, and best regards

G. Bianchini

**Detailed list of corrections:**

*P3,L9: replace "problematics" with "challenges"*

corrected

*Fig 1 caption: left and right panels, not top and bottom*

I think in the final 2-column form this caption will be correct, while in the discussion format the images are put side by side

*P5,L11: replace "do" with "does"*
*P7,L24: replace "it is shown the result of this analysis for two different lines" with "the results for two different lines are shown"*
*P11,L1: "provided an Ethernet connection"*
*P12,L3: "expected atmospheric radiance signal"*
*P12,L4: remove "with respect to the instrument accuracy"*

corrected

*P12,L17: "continuously performed (or when very dry conditions exist which is frequently at Concordia) during a multi-year"*

rephrased, actually "continuously" is not correct (it is not necessary to perform this test continuously), so I specified that this kind of calibration "can be performed whenever needed, during a multi-year deployment of the instrument, provided very dry atmospheric conditions are present (which is frequently the case for Dome C)"

*Fig 8: the gray line showing the difference is very hard to see; please increase the context*

*P13,L6: "of the system; however, this is at the cost of a lower absolute stability of the spectral calibration reference."*
*Pg 14,L3: replace "has not to be" with "does not need to be"*
*P15,L6: replace "performances" with "performance"*
*P18,L1: replace "evidence" with either "importance" or "note"*

corrected

*Pg18,L10 (and Fig 13): the downwelling radiance spectrum in the right-hand panel is not from a tropical atmosphere. The tropics have a large PWV and thus there will be significant downwelling radiance in the 800-1000 cm-1 spectral region (not ~zero as shown here). My guess is that the wrong spectrum was accidentally plotted here*

Actually the ground-based spectrum is from a tropical-equatorial region, but from an high altitude site (Cerro Toco, 5500 m a.s.l., measurement taken during the RHUBC-II campaign): this has been clarified better in the text

*Pg 18,last line: "a software package has"*
*Pg19,L3: "The software retrieved temperature and water vapor content profiles"*
*Eq1: no "+" in the equation*
*Pg 19,L11: "where y and x are the vector" (they were accidentally reversed in the paper)*
*Pg 19,L12: "VCM" has not been defined in this paper*

corrected

*Pg20,L10: "Figure 13 left panel"*
*Fig 15 caption: "in Figure 13 left panel"*

same as above: figure positioning will change with 2-column format

*Pg20,L21: "data product. The accuracy in the"*

corrected

*Fig 16: Question: is the low level warm temperatures in the right panel (lowest few m) actually the temperature inside the instrument and/or chimney? If so, this should be noted in the caption or text.*

Yes, added few words to clarify both in text and caption

*Pg21,L4: "be retrieved. For example, nitrous oxide is obtained…"*
*Pg21,L6: "In Figurew 17 bottom panel a time series of the retrieved N2O*

*obtained …period is shown."*
*Pg21,L9: "…is added to the retrieval range. However, the main methane absorption feature overlaps with the absorption bands of the Mylar…" (i.e., remove the "(differently from…main absorption features"*
*Fig 17 caption: "OMI measurements over the Dome C region are also shown."*

corrected

*Pg 22,L9: is there a reference for the OMI data you are using?*

I used the public web interface provided by NOAA, I added the URL as a footnote in the text

*Pg22,L16: "analysis process provides a valuable tool"*
*Pg22,L17: "Figure 18 shows the result…"*
*P22,L19: "without any significant maintanence."*
*P23,L20: "relevant cloud-free atmosphere processes, which is the reference case in this work."*

corrected

*P24,L13: The extended-range AERI at the ARM NSA site, which has been operating there since 1999, is a similar instrument to the REFIR-PAD in that it measures downwelling spectral radiance in the mid and (part of the) far-IR (although not as deeply into the far-IR). So the statement that "no similar instruments are operating continuously in polar regions" is mis-leading, and should either be removed or reworded.*

I made a wrong use of the term "polar", actually I was meaning "Antarctic", so I corrected accordingly the text